# A Novel Interaction of Slug (SNAI2) and Nuclear Actin

**DOI:** 10.3390/cells13080696

**Published:** 2024-04-17

**Authors:** Ling Zhuo, Jan B. Stöckl, Thomas Fröhlich, Simone Moser, Angelika M. Vollmar, Stefan Zahler

**Affiliations:** 1Center for Drug Research, Ludwig-Maximilians-University Munich, Butenandtstr, 5-13, 81377 Munich, Germany; ling.zhuo@cup.uni-muenchen.de (L.Z.); angelika.vollmar@cup.uni-muenchen.de (A.M.V.); 2Laboratory for Functional Genome Analysis, Gene Center Munich, Ludwig-Maximilians-University Munich, Feodor-Lynen-Str. 25, 81377 Munich, Germany; stoeckl@genzentrum.lmu.de (J.B.S.); frohlich@genzentrum.lmu.de (T.F.); 3Department of Pharmacognosy, Institute of Pharmacy, University of Innsbruck, Innrain 80/82, 6020 Innsbruck, Austria; simone.moser@uibk.ac.at

**Keywords:** nuclear actin, slug, SNAI2, DNA damage, apoptosis

## Abstract

Actin is a protein of central importance to many cellular functions. Its localization and activity are regulated by interactions with a high number of actin-binding proteins. In a yeast two-hybrid (Y2H) screening system, snail family transcriptional repressor 2 (SNAI2 or slug) was identified as a yet unknown potential actin-binding protein. We validated this interaction using immunoprecipitation and analyzed the functional relation between slug and actin. Since both proteins have been reported to be involved in DNA double-strand break (DSB) repair, we focused on their interaction during this process after treatment with doxorubicin or UV irradiation. Confocal microscopy elicits that the overexpression of actin fused to an NLS stabilizes complexes of slug and γH2AX, an early marker of DNA damage repair.

## 1. Introduction

Actin exerts multiple key functions in eukaryotic cells, among which definition of the cell shape and enabling cell motility have been known for the longest time. Actin is one of the most abundant proteins in cells and is ubiquitously expressed; hence, its functions are temporally and spatially tightly regulated by interaction with actin-binding proteins (ABPs), which can either cause nucleation, fragmentation, polymerization, branching, or bundling of the actin filaments [1]. With increasing knowledge about the structural basis of these interactions [2], it became feasible for the protein–protein interactions between actin and its binding proteins to be pharmacologically addressed using small molecules [3,4,5]. Research from recent years has revealed that actin and several actin-binding proteins also exist in the nucleus [6,7]. Therefore, actin has been shown to participate in transcriptional regulation, as well as in DNA damage repair [7,8,9,10].

Slug, also known as SNAI2, is a protein that plays a crucial role in embryonic development and tissue formation. It belongs to the Snail family of transcription factors, which are involved in regulating gene expression during development. Slug is primarily known for its role in the epithelial–mesenchymal transition (EMT), a process in which epithelial cells acquire mesenchymal characteristics. The EMT is essential to embryonic development and wound healing, but it can also contribute to cancer metastasis when deregulated [11]. Furthermore, slug plays an important role during the DNA damage response and apoptosis [12,13,14].

Here, we describe the identification of an interaction between actin and slug and characterize its impact on the DNA damage response after the treatment of HeLa cells with doxorubicin or UV irradiation.

## 2. Materials and Methods

### 2.1. The Yeast Two-Hybrid System

The coding sequence of beta actin was amplified from an actin-containing plasmid for mammalian expression, pCAG-mGFP-Actin, which was a gift from Ryohei Yasuda (Addgene plasmid # 21948; URL (accessed on 15 April 2024, http://n2t.net/addgene:21948; RRID:Addgene_21948) [15], using the following primers, adding BamHI and EcoRI restriction sites: Forward primer: 5′-C ATG GAG GCC GAATTC ATG GAT GAT GAT ATC GCC GCG CTC-3′.

Reverse primer: 5′-GC AGGTCGACGGATCC GAA GCA TTT GCG GTG GAC GAT GGA-3′.

The PCR product was analyzed using a 1% agarose gel and purified using a gel extraction kit (QIAGEN, Hilden, Germany). The purified insert and the bait vector pGBKT7 (Takara Bio, Saint-Germain-en-Laye, France) were digested with BamHI and EcoRI (Thermo Fisher Scientific, Darmstadt, Germany). The vector was further dephosphorylated by directly adding 1 µL of FastAP Thermosensitive Alkaline Phosphatase (Thermo Scientific, Darmstadt, Germany). The vector and insert were again purified using a gel extraction kit. Ligation was carried out at a 1:3 ratio (vector: insert) using the protocol from Thermo Fisher Scientific at room temperature for 30 min, followed by heat shock transformation of the ligation mixture into the chemical competent *E. coli* DH5-α. Plasmid isolation was performed using the QIAprep^®^ Spin Miniprep Kit (QIAGEN, Hilden, Germany). The plasmids were analyzed using PCR and their sequencing using a standard T7 primer (Eurofins, Ebersberg, Germany).

The Matchmaker Gold yeast two-hybrid system (Takara Bio) was used to screen for the interaction partners of actin following the manufacturer’s instructions. In brief, the vector was introduced into the Y2H Gold reporter strain, which was then mated with the Mate&Plate^TM^ library—Universal Human (normalized) (Takara Bio) in yeast strain Y187. The screening was performed following the Matchmaker manual. An aliquot of 4.7 × 10^7^ cfu was screened (5-fold library coverage) on plates (15 cm diameter) containing DDO/X/A medium. After incubation, 960 blue colonies were picked from the plates and re-spotted onto 10 15 cm diameter plates each containing the following media (40 plates in total): DDO, DDO/X/A, QDO, and QDO/X/A. A total of 480 colonies which showed robust growth on all plates and a blue color on both plates containing X-α-Gal were classified as “hits” and were spotted onto the DDO plates to ensure robust growth. Then, colony PCR was performed for those colonies. Samples showing a band (even if it was weak) on agarose gel were analyzed using sequencing (76 samples with Eurofins Genomics, 464 samples with Macrogen Europe, both T7 standard sequencing primer; some samples were repeated due to bad sequencing quality). The identity of the insert was revealed using a BLAST search (URL accessed on 15 April 2024, https://blast.ncbi.nlm.nih.gov/). SNAI2 was identified once. The colony containing the SNAI2 hit was selected, and the plasmid was isolated (Zymoprep™ Yeast Plasmid Miniprep kit, Zymo Research, Freiburg, Germany).

### 2.2. Cell Culture

The HeLa cells in this study were from DSMZ (Leibniz Institute, Braunschweig, Germany). The cells were cultured in DMEM supplemented with 10% fetal bovine serum (FBS) and incubated at 37 °C and 95% humidity with 5% CO_2_. The cells were either treated with doxorubicin or a matched concentration of DMSO as the solvent control, as indicated in the figures. For irradiation with UV, the cells were illuminated at 365 nm for 60 s at an intensity of 100% (corresponding to approximately 23.1 mW/cm^2^) with a benchtop illuminator from Rapp OptoElectronic (Wedel, Germany).

### 2.3. Apoptosis Assay

Apoptosis was measured according to Nicoletti et al. [16]. Briefly, the harvested cells, medium, and washing solution were all transferred into flow cytometry tubes and centrifuged with 600× *g* for 10 min at 4 °C. Debris or contaminants were washed away twice. The cells were resuspended in Nicoletti buffer (0.1% sodium citrate (*w*/*v*) + 0.1% Triton X-100 (*v*/*v*) in PBS) and incubated with propidium iodide (50 µg/mL) for 1 h. The fluorescence intensity of propidium iodide in the cells was measured using flow cytometry using a FACSCanto II (Becton Dickinson, Heidelberg, Germany).

### 2.4. Immunoprecipitation

Cells in 6-well dishes were washed with ice-cold PBS 3 times, and then 100 µL of RIPA lysis buffer was added to allow for lysis of both the plasma membrane and nuclear membrane. The plate was incubated on ice for 30 min with occasional shaking. The cells were scraped off and transferred into 1.5 mL tubes. The cell debris was removed using high-speed centrifugation (10,000× *g*, 10 min, 4 °C); the supernatant was collected. The protein concentration was determined according to Bradford assay. A total of 2 μg (1:50) of monoclonal antibody was mixed with the supernatant, and 50 µL of μMACS Protein G MicroBeads (Miltenyi Biotec, Bergisch Gladbach, Germany) was added overnight at 4 °C with shaking. A μColumn (Miltenyi Biotec, Bergisch Gladbach, Germany) was rinsed with 200 µL of lysis buffer, and the lysate was allowed to run through the column. After incubation with the protein solution, the column was rinsed 4× with 200 µL of RIPA buffer and with 100 µL of low-salt buffer once. After rinsing, the column was eluted with 20 µL of pre-heated SDS-PAGE sample buffer and incubated for 5 min at room temperature. Another 50 µL of pre-heated SDS-PAGE sample buffer was eluted in a fresh 1.5 mL tube. The sample was boiled for 5 min at 95 °C and stored at −20 °C for subsequent Western blot analysis or mass spectrometry. For the IP experiments with nuclear actin, mCherry-labeled actin with multiple nuclear localization sequences (see the section on transfection) was overexpressed and precipitated with an anti-mCherry antibody to ensure that primarily nuclear actin was enriched.

### 2.5. Western Blot

Equal protein amounts of the denatured samples were subjected to SDS-PAGE on 10% polyacrylamide gel. Electrophoresis was performed at 100 V for 20 min before the actual separation at 200 V for 45 min. The total protein amount of each lane was quantified using stain-free technology, and the molecular weight of the bands was assessed through comparison with a PageRuler Plus Prestained Protein Ladder. The separated proteins were then transferred from the gel onto a PVDF membrane (0.45 μm pore size) at 100 V for 90 min at 4 °C. Non-specific binding proteins were blocked with 5% Blotto-TBST for 2 h, and the membrane was incubated in TBST containing primary antibody (1:1000) overnight at 4 °C with gentle agitation. As the primary antibodies, a monoclonal Anti-ACTB antibody from Sigma-Aldrich (Darmstadt, Germany) and an anti-slug (C19G7) Rabbit mAb from Cell Signaling Technology (Danvers, MA, USA) were used. On the next day, the membrane was washed with 0.1% Tween-20 in TBST for 5 min 4 times, following incubation with a suitable secondary HRP-conjugated antibody for 2 h at room temperature with gentle shaking. To remove the unbound secondary antibody, the membranes were washed with 0.1% Tween-20 in TBST for 4 × 5 min each. After brief incubation with HRP Homemade ECL solution, the membrane was imaged using a ChemiDoc Touch Imaging System. The equal loading and homogenous transfer of the proteins were evaluated by measuring their band intensities using ImageLab and normalizing them to the total protein amount (stain-free gel) as the loading control.

### 2.6. Transfection

The cells were grown to 80% confluence in a 6-well plate. For each group, 1 µg of plasmid DNA was mixed with the Lipofectamine™ 3000 reagent (Thermo Fisher Scientific, Darmstadt, Germany), according to the manufacturer’s instructions. The Nuclear Actin-Chromobody^®^ plasmid was from Chromotek (Planegg, Germany). pmCherry-C1 actin-3XNLS P2A mCherry was a gift from Dyche Mullins (Addgene plasmid # 58475; URL accessed on 15 April 2024, http://n2t.net/addgene:58475; RRID:Addgene_58475) [17]. For the knockdown of slug, the DharmaFECT siRNA transfection protocol (Horizon Discovery, Cambridge, UK) was used. A total of 300,000 cells/well were seeded in a 12-well plate. A total amount of 5 µM of SNAI2 (6591) siRNA was added to the DharmaFECT 1 Transfection Reagent complex and incubated for 24 h. The transfected cells were seeded on ibidi 8-well µ-slides (45,000 cells/well) for further experiments.

### 2.7. Confocal Microscopy

Cells on Ibitreat^®^ (Ibidi, Gräfelfing, Germany) 8-well µ-slides were washed with PBS^+^/Ca^2+^/Mg^2+^ and fixed with 4% formaldehyde for 15 min at room temperature. The fixed cells were rinsed with PBS for 10 min of shaking, permeabilized in 0.2% Triton-100 in PBS for 30 min, and washed with PBS for 10 min of shaking. After blocking them with 1% BSA in PBS for another 30 min at RT, the cells were incubated with primary antibody diluted in 0.2% BSA (1:200) overnight at 4 °C. The slides were then washed with 1% BSA in PBS for 3 × 10 min and then incubated with the secondary antibody (1:400) and Hoechst 33342 in PBS for 1 h at RT. Finally, the cells were washed for 2 × 10 min with 1% BSA in PBS and once with PBS for 10 min. The slides were sealed with FluorSave Reagent and stored at 4 °C in the dark. The cells were stained for F-actin with rhodamin–phalloidin (Invitrogen, Darmstadt, Germany) for 1 h and then washed for 3 × 10 min with PBS at RT. Images were acquired using a TCS SP8 SMD inverted confocal microscope (Leica, Mannheim, Germany) with a HC PL APO 40×/1.30 OIL or 63×/1.40 OIL immersion objective lens. Scanning was performed at 400 Hz, and an average of four frames was acquired for the colocalization analysis. The following excitation laser lines were used: 405 nm, 488 nm, 552 nm, and 638 nm. The number of actin aggregates was counted using ImageJ version 1.53t. The particles were analyzed with the settings of size: 0.03–5.00 and circularity: 0.2–1.00. Colocalization was calculated using the respective plugin from ImageJ. The primary antibodies were rabbit anti-slug (C19G7) from Cell Signaling Technology (Danvers, MA, USA), mouse anti-RPA32/RPA2 (ab2175) from Abcam (Cambridge, UK), and mouse anti-Phospho-Histone H2A.X (80312) from Cell Signaling Technology. The secondary antibodies were goat anti-rabbit or goat anti-mouse labeled with AlexaFluor 488, 546, 633, or 680, respectively (Invitrogen, Darmstadt, Germany).

### 2.8. Liquid Chromatography–Mass Spectrometry Analysis

Prior to the mass spectrometry analysis, the beads from the immunoprecipitation experiments were subjected to tryptic on-bead protein digestion. For this, the beads were incubated overnight at 37 °C in 100 µL of trypsin solution (500 ng modified porcine trypsin (Promega, Madison, WI, USA), 1 M urea, 400 mM NH_4_HCO_3_). The supernatant was collected, and the peptides were further extracted from the beads in two 10–15 min wash steps with 50 µL of 50 mM NH_4_HCO_3_. The peptides in the pooled fractions were first reduced for 10 min using dithioerythritol (concentration: 1 mM) and then carbamidomethylated with iodoacetamide (concentration: 1.35 mM). After 30 min of incubation, the digestion was stopped according to the addition of 2.7 µL of trifluoroacetic acid. Then, 10% of each peptide sample was analyzed using liquid chromatography–tandem mass spectrometry (LC-MS/MS) using an Ultimate 3000 nano-LC system coupled to a Q Exactive HF-X Orbitrap mass spectrometer (Thermo Fisher Scientific, Darmstadt, Germany). The samples were injected onto a trap column (PepMap 100 C18, 100 μm × 2 cm, 5 μM particles, Thermo Fisher Scientific) and separated at 250 nL/min using an EASY-Spray column (PepMap RSLC C18, 75 μm × 50 cm, 2 μm particles, Thermo Fisher Scientific). As mobile phase A, 99.9/0.1% water/formic acid (*v*/*v*), and as mobile phase B, 99.9/0.1% acetonitrile/formic acid (*v*/*v*) were used. The chromatography method included an 80 min gradient from 5% to 20% B, followed by an increase to 40% in 9 min. The MS acquisition was performed in the data-independent mode with 12 *m*/*z*-wide windows (400–1000 *m*/*z*, 15,000 resolution, NCE 27) using a staggered window pattern. For library-free protein identification and quantification, DIA-NN (v1.8.1) [18], in combination with the human subset of the Swiss-Prot database, was used and the deep-learning-based spectra and retention time prediction feature activated. The false discovery rate was set to 1%. Statistical evaluation was carried out using Perseus [19]. For comparison of the control conditions with the conditions directly after UV irradiation, Welch’s *t*-test was performed and corrected for multiple testing using a permutation-based approach. Only proteins altered by at least 30% were considered significantly differentially abundant. For the comparison of different conditions, one-way ANOVA was performed and corrected for multiple testing. Significant proteins were subsequently analyzed using Tukey’s HSD test. The false discovery rate for all the quantitative comparisons was controlled to be 5%.

### 2.9. Statistics

Three independent replicates were performed for each experiment. The protein level of the Western blot results was normalized to the total protein amount and quantified using Image Lab™ software 6.1 (Bio-Rad, Munich, Germany). The data on the Western blot, confocal imaging, and flow cytometry were normalized to the vehicle control (DMSO). All the confocal images were evaluated using ImageJ version 1.53t. The data in bar graphs are presented as the mean values or mean ± standard error of the mean (SEM). The statistical analysis and data plotting were performed using Prism Version 9.5.1 (GraphPad Software, Boston, MA, USA). The statistical methods and significances are indicated in the respective figures.

The potential synergism of the combination treatment was assessed through calculation of the Bliss value [20,21].

## 3. Results

### 3.1. Actin Interacts with Slug

A yeast two-hybrid approach with actin as the bait elicited several unknown interacting proteins in addition to already known actin binders, e.g., cofilin 2 or CAP2 (Table 1).

From these novel putative actin-binding proteins, we selected slug (SNAI2) for further studies and validated its binding to actin via co-immunoprecipitation (Figure 1).

### 3.2. Slug and Actin Mutually Influence Each Other

The silencing of slug by siRNA led to a rearrangement of the organization of the cytoplasmic F-actin fibers and to the emergence of aggregates (Figure 2a). Since slug is mainly present in the nucleus, there obviously is some feedback from the nuclear compartment to the cytoplasm, indicating the functional importance of the interaction of actin and slug.

This view is supported by the observation that the treatment of cells with actin-depolymerizing or -stabilizing compounds (latrunculin B and jasplakinolide, respectively) causes changes in the overall levels of slug or the intracellular distribution of slug (Figure 2b,c).

### 3.3. The Manipulation of Actin Acts Synergistically with DNA Damage by Doxorubicin or UVA Irradiation

We have previously shown that the treatment of cells with the actin-polymerizing compound chondramide B acts synergistically with the DNA-damaging agent doxorubicin and that this action is based on the inhibition of DNA repair [22]. Since slug has also been described as relevant in this context [12,13,14], we first tried to reproduce the effect of actin’s manipulation on the response to DNA damage in the HeLa cells. Pretreatment of the cells with increasing concentrations of both Lat B (Figure 3a) and jasp (Figure 3b), caused a synergistic increase in apoptosis after co-treatment with doxorubicin (doxo), as well as after irradiation with UVA (Figure 3c).

### 3.4. Slug and Actin Fused with an NLS Show Weak Spatial Correlation during DNA Damage Repair

Since there are only very low levels of endogenous nuclear actin, we studied the colocalization of slug with nuclear actin after the overexpression of anti-actin nanobodies or actin, both with nuclear localization sequences. In both cases, we observed a weak colocalization of slug and actin in the nucleus throughout the observed DNA repair time of 5 h (Figure 4), indicated by a Pearson’s correlation coefficient of around 0.3.

### 3.5. The Colocalization of Slug with Other Proteins of DNA Repair Complexes Changes during Repair and Depending on the Levels of Actin with an NLS

We recently showed the important role of the interaction of RPA2 with nuclear actin during DNA damage repair [22], and the importance of slug to the stability of γH2AX foci has also been demonstrated previously [23]. Therefore, we studied the kinetics of the colocalization between slug and γH2AX, as well as between slug and RPA2, during DNA repair, both under control conditions and after increasing the nuclear actin levels. Interestingly, the colocalization between slug and γH2AX increased in response to DNA damage (Figure 5a). This increase was even more pronounced after increasing the levels of nuclear actin (Figure 5a). The colocalization of slug and RPA2 was only slightly and transiently increased after DNA damage, but also this spatial correlation seems to be stabilized by nuclear actin (Figure 5b).

Analogous to the induction of DNA damage with doxorubicin, we also investigated the colocalization between slug and γH2AX after irradiation with UV light. The colocalization increased during DNA repair (Figure 6). This effect was not altered by the presence of latrunculin B, an inhibitor of actin polymerization (Figure 6).

To investigate the interactions of nuclear actin with other proteins during DNA damage repair in an unbiased manner, we performed pulldown experiments with overexpressed actin fused with an NLS and subsequent analysis using mass spectrometry. We first compared the proteins interacting with nuclear actin under control conditions and immediately after irradiation with UV. Interestingly, several proteins which have previously been described as either important to DNA damage repair (e.g., NUDT3, SPTBN1) or interactors of actin (SCD5, SPTAN1, SPTBN1) were in the group of proteins influenced by UV irradiation (Table 2).

Multiple comparisons of the proteins interacting with nuclear actin were performed under different experimental conditions, including a 1 h repair time after UV-induced damage or the addition of the actin-depolymerizing agent Lat B. Though several proteins significantly changed in their interaction with actin, we found no further systematic changes in the interactors under these conditions (Appendix A).

## 4. Discussion

Actin is one of the most abundant proteins and one of the proteins with the most known interactors in mammalian cells. Approaches to identifying protein–protein interactions are limited due to the potentially transitory and weak binding of the interaction partners. Here, we describe a yeast two-hybrid approach, which elicited the transcriptional regulator slug (SNAI2) as a novel interactor with actin. Since this interaction was independently validated using immunoprecipitation, we explored the functional consequences of this finding. Overall, silencing of slug caused alterations in the cytoplasmic actin structure, and the use of actin-manipulating small molecules (jasplakinolide, latrunculin) caused changes in slug expression. This indicates that the interaction is functionally meaningful since there seems to be crosswise regulation [24], which has previously been hypothesized to occur via the MRTF pathway [25]. Since, as a transcriptional regulator, slug is mainly present in the cell nucleus, we then focused on the nuclear compartment. We [22] and others have previously demonstrated the importance of nuclear actin to the repair of DNA damage [7,9,26]. In good accordance with previous results [22], we validated the synergistic effect of manipulating the actin dynamics via small molecules and treatment with either a DNA-damaging compound (doxorubicin) or UV irradiation. Indeed, for slug, an important role during DNA damage repair and apoptosis induction has been postulated [12,13,23]. To investigate the putative interaction of slug and nuclear actin, we either overexpressed fluorescence-labeled actin with multiple nuclear localization sequences or a nuclear chromobody for actin. Both are accepted tools for studying nuclear actin [27]. The direct interaction between actin and slug seemed to be moderate and not regulated during DNA repair in our experimental model. In contrast, the interaction between γH2AX and slug increases during DNA damage repair and seems to be stabilized by the expression of actin fused with an NLS. It has recently been shown that silencing of slug stabilizes γH2AX foci and thus damage resolution. One could speculate that a ternary complex consisting of nuclear actin, slug, and γH2AX is formed during DNA damage repair. Manipulating this complex by either silencing slug or changing the actin architecture in the nucleus could lead to aberrant stabilization of the damage repair foci. A similar effect also occurred after DNA damage due to UV irradiation. Interestingly, the interaction between slug and γH2AX was not altered by treatment with Lat B. In a previous work [22], we demonstrated that Lat B is able to release the interaction of nuclear actin and RPA2. Thus, the effect we observe in the present work seems to have a different mechanistic background. It should, however, be kept in mind that the overexpression of actin fused with an NLS or a nuclear chromobody against actin might shift the balance between nuclear F- and G-actin and might thus bias the functional outcome for DNA damage repair.

In order to gain insight into the changing interaction partners of nuclear actin during DNA damage and repair, we performed the overexpression of actin fused with an NLS, followed by immunoprecipitation and proteomic analysis of the precipitate. Unfortunately, none of the classic DNA damage repair proteins were detectable using mass spectrometry. This was most likely due to the low abundance of these proteins or their putatively low binding affinity to actin. This might also be why we did not detect slug following this approach. However, the list of proteins which show altered interaction with nuclear actin after irradiation still gives some interesting hints towards yet unidentified interactions during DNA damage repair. NUDT3, which has previously been shown to limit DNA damage [28], interacted significantly less with nuclear actin after UV irradiation. Likewise, SCD5, which is known as an actin-organizing protein in yeast [29], and ZMIZ1, which is a transcriptional co-activator of p53 [30], the “guardian of the genome”, interacted less with nuclear actin upon irradiation with UV light. In contrast, SPTAN1 (also known as alpha spectrin II) and SPTBN1 (beta spectrin II) both interacted more with nuclear actin after UV irradiation. Both proteins are known as actin binders and have previously been identified as important to DNA damage repair [31,32]. Interestingly, the depletion of SPTBN1 has been shown to cause nuclear accumulation of slug [33].

In conclusion, we identified a novel interaction between slug and actin, which seems to play a role during DNA damage repair. This interaction seems to be complex and to involve other proteins, like γH2AX. Our work suggests that nuclear actin orchestrates the DNA damage response in more ways than previously assumed.

## Figures and Tables

**Figure 1 cells-13-00696-f001:**
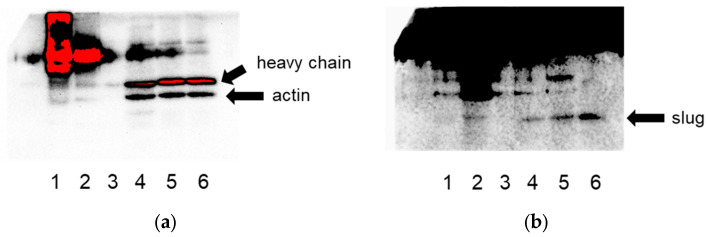
Co-immunoprecipitation of slug and actin in both directions in lysates of HeLa cells confirms their interaction. (**a**) Detection for actin, (**b**) detection for slug. Lane 1: IP for actin, lane 2: IP for slug, lane 3: negative control (beads), lanes 4–6: the respective flow-throughs.

**Figure 2 cells-13-00696-f002:**
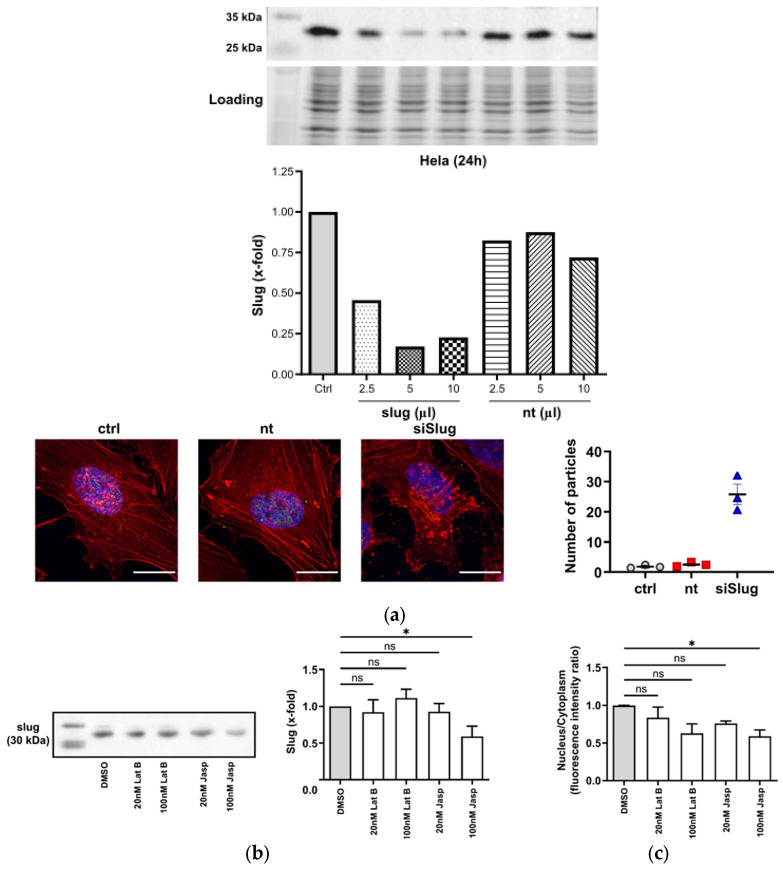
Slug and actin mutually influence each other. (**a**) Upper panel: Slug is successfully silenced to 20% of the initial level 24 h after treatment with 5 µL of transfection solution. Lower-left panel: After silencing of slug, actin aggregates emerge; red: F-actin, blue: nuclei, scale bar: 25 µm. Lower-right panel: Quantitative analysis of the actin aggregates. Nt: non-targeting siRNA. The aggregation of F-actin was quantified by counting the number of particles in 40 single cells (from duplicate wells of three independent experiments); n = 3, mean ± SEM. (**b**) Left panel: Representative Western blot showing the slug level after the addition of the indicated concentrations of latrunculin B (Lat B) and jasplakinolide (jasp) for 24 h. The leftmost lane depicts the size marker. Right panel: Quantitative densitometric analysis of the Western blots showing a decrease in slug levels after treatment with 100 nM of the actin-polymerizing compound jasp. Protein levels were normalized to solvent control. (n = 3, mean ± SEM, half-tailed unpaired equal-variance *t*-test * *p* < 0.05). (**c**) Nuclear–cytoplasmic ratio of slug after the treatment panels analogous to (**b**). The nuclear fluorescence intensity of slug as determined using confocal microscopy was reduced after 24 h treatment. The N/C ratios were estimated using Fiji software 2.9.0 and normalized to DMSO control. (n = 3, mean ± SEM, half-tailed unpaired equal-variance *t*-test * *p* < 0.05, ns: not significant).

**Figure 3 cells-13-00696-f003:**
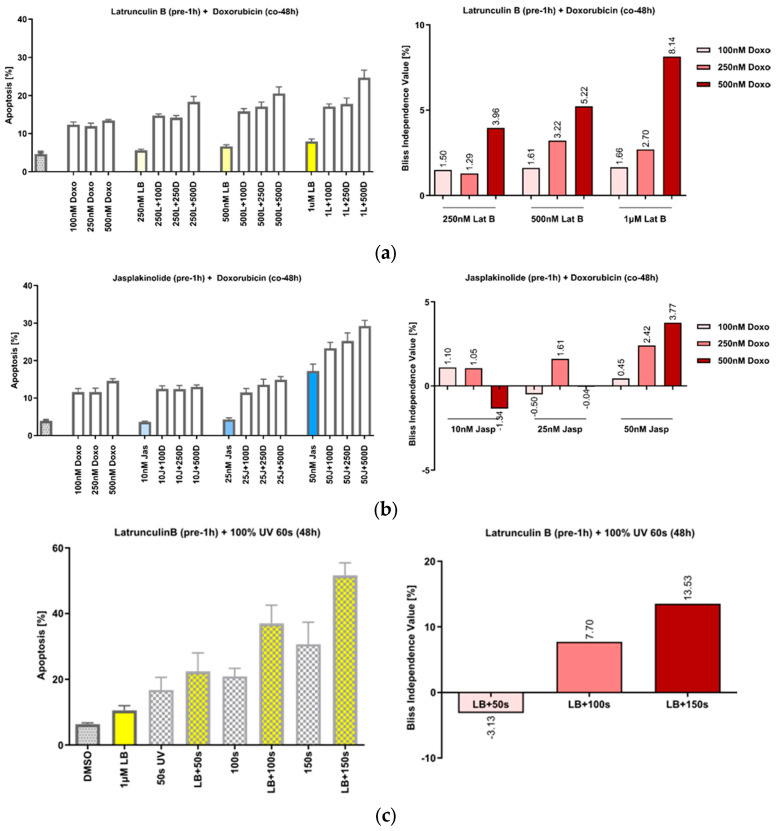
Actin-manipulating compounds and DNA-damaging interventions have a synergistic effect on apoptosis induction in HeLa cells. (**a**) Left panel: Effect of the co-treatment of cells with increasing concentrations of Lat B and doxorubicin; right panel: the calculated Bliss values show actin-binding compounds and doxo dose-dependently increased apoptosis in HeLa. (**b**) Co-treatment of the cells with jasp, panels analogous to (**a**). (**c**) Treatment of the cells with UVA irradiation, panels analogous to (**a**). n = 3, mean ± SEM. The Bliss independence value is often used to analyze synergistic effects of drug combinations. Values between 0 and 1 represent antagonism; values higher than 1 represent synergism.

**Figure 4 cells-13-00696-f004:**
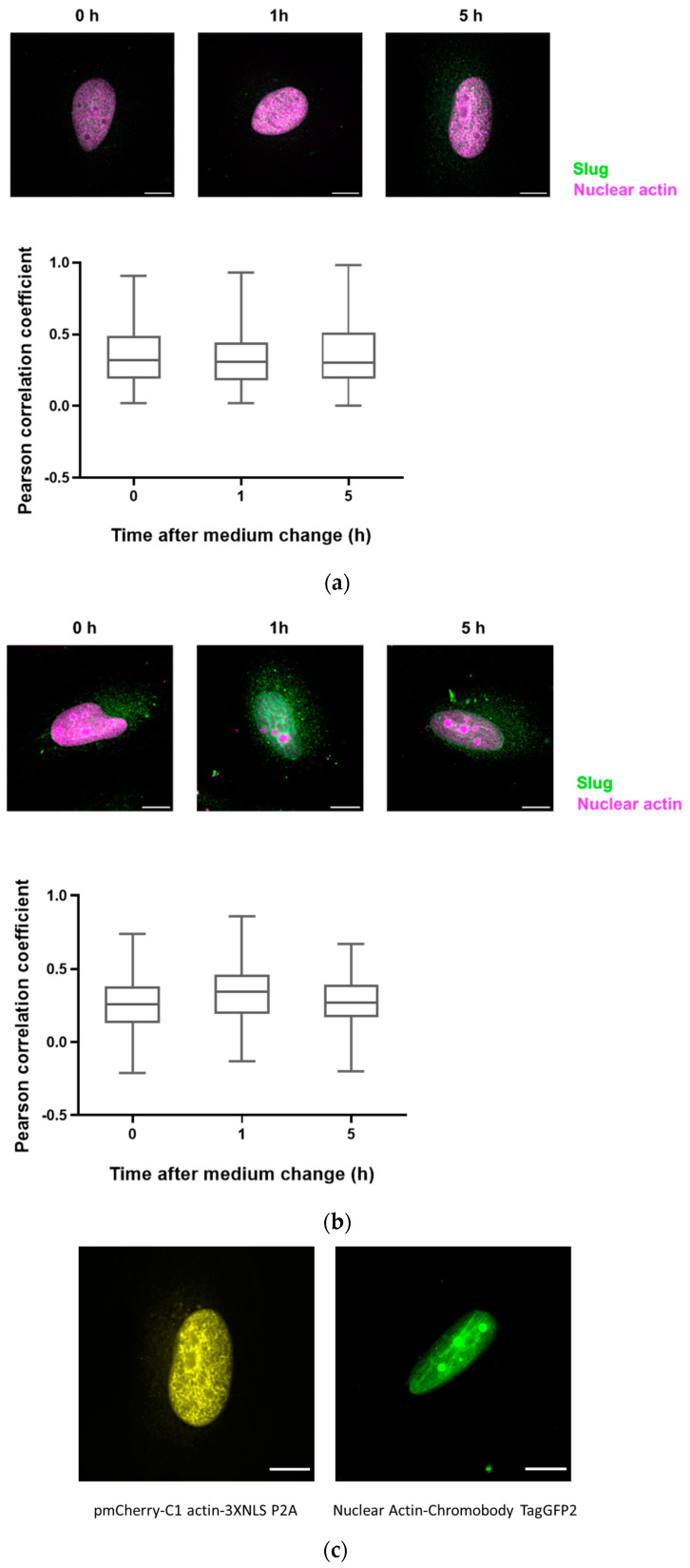
Slug and actin fused with NLS colocalize during the DNA damage repair process. (**a**) The cells were overexpressed with actin fused with mCherry and NLS. Slug was stained using indirect immunofluorescence (r = 0.35 ± 0.03). (**b**) The cells were overexpressed with nuclear Actin-Chromobody (TagGFP2) (r = 0.30 ± 0.02). n = 3, mean ± SEM, scale bar: 10 μm, single cell number for each group: 20, number of square regions of each nucleus for evaluation: 3. (**c**) Both methods for increasing nuclear actin elicited actin fibers in the nucleus, scale bar: 10 μm.

**Figure 5 cells-13-00696-f005:**
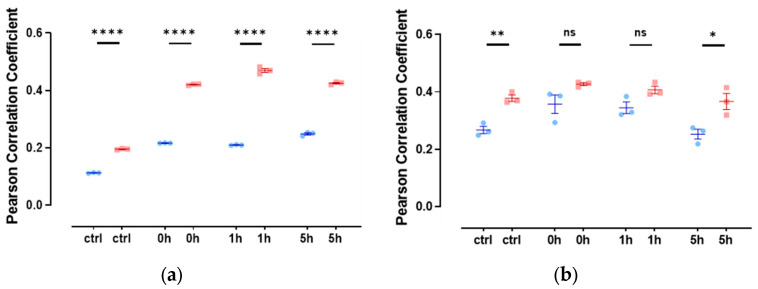
The colocalization between slug and γH2AX, as well as between slug and RPA2, is stabilized by nuclear actin. (**a**) Colocalization between slug and γH2AX during DNA repair with (red symbols) and without (blue symbols) expression of nuclear actin; (**b**) colocalization between slug and RPA2 during DNA repair with (red symbols) and without (blue symbols) expression of nuclear actin. n = 3, mean ± SEM, single cell number for each group: 20, number of square regions of each nucleus for evaluation: 3. * *p* < 0.05, ** *p* < 0.01, **** *p* < 0.0001, ns: not significant.

**Figure 6 cells-13-00696-f006:**
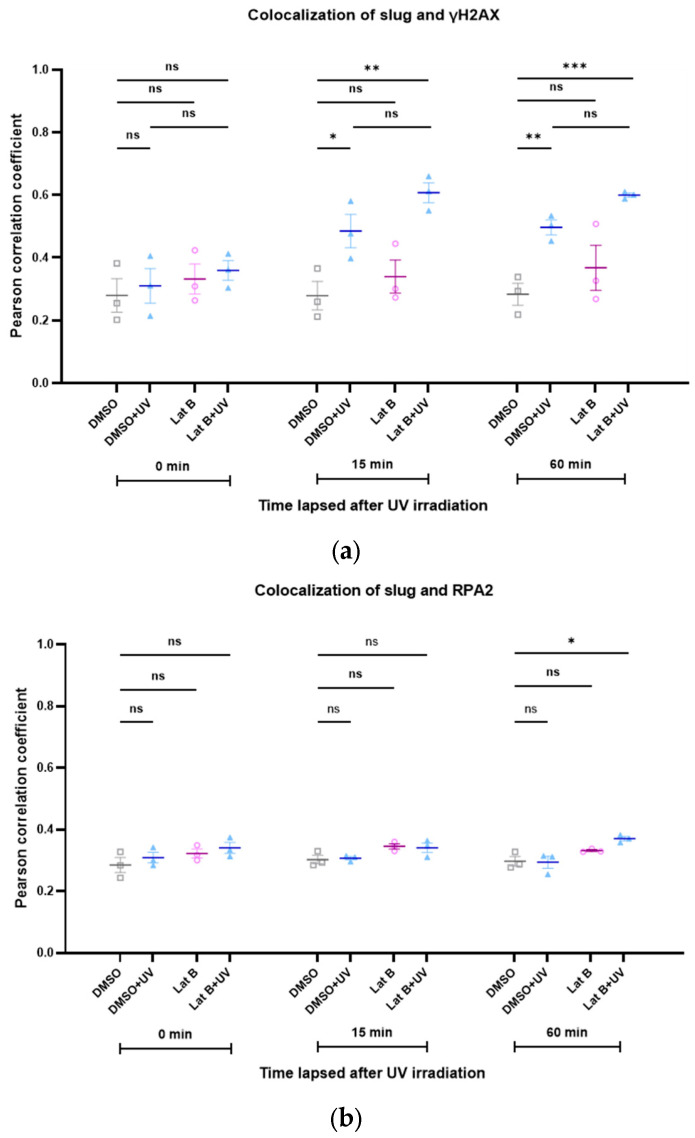
(**a**) The spatial correlation between slug and γH2AX increases after induction of DNA damage with UV irradiation. Latrunculin B does not influence this effect. (**b**) The spatial correlation between slug, RPA2, and γH2AX is not changed after induction of DNA damage with UV irradiation. n = 3, mean ± SEM, single cell number for each group: 20, number of square regions of each nucleus for evaluation: 3. ns: not significant, * *p* < 0.05, ** *p* < 0.01, *** *p* < 0.001.

**Table 1 cells-13-00696-t001:** Proteins detected as actin interactors via the yeast two-hybrid approach.

Insert Identity	Frequency
3′-phosphoadenosine 5′-phosphosulfate synthase 1 (PAPSS1)	1
BRX1, biogenesis of ribosomes (BRIX1)	1
chromosome 9 open reading frame 131 (C9orf131)	1
cofilin 2 (CFL2)	16
COP9 signalosome subunit 5 (COPS5)	8
cyclase-associated actin cytoskeleton regulatory protein 2 (CAP2)	164
cysteine sulfinic acid decarboxylase (CSAD)	3
double zinc ribbon and ankyrin repeat domains 1 (DZANK1)	1
dual-specificity phosphatase 5 (DUSP5)	3
endoplasmic reticulum lectin 1 (ERLEC1)	1
ENY2, transcription and export complex 2 subunit (ENY2)	1
epidermal growth factor receptor pathway substrate 8 (EPS8)	1
F-box and WD repeat domain-containing 7 (FBXW7)	1
gamma-aminobutyric acid type A receptor alpha4 subunit (GABRA4)	1
integrin subunit alpha 8 (ITGA8)	2
late endosomal/lysosomal adaptor, MAPK and MTOR activator 5 (LAMTOR5)	8
lysine acetyltransferase 14 (KAT14)	10
NLR family pyrin domain-containing 1 (NLRP1)	13
nuclear receptor binding factor 2 (NRBF2)	2
phosphodiesterase 1C (PDE1C)	1
RAD51 associated protein 2 (RAD51AP2)	62
ribosomal protein L26 (RPL26)	1
snail family transcriptional repressor 2 (SNAI2)	1
SNAP-associated protein (SNAPIN)	9
translation machinery associated 7 homolog (TMA7)	11
VHL-binding protein 1 (VBP1)	3
zinc finger CCHC-type containing 4 (ZCCHC4)	2
zinc finger protein 148 (ZNF148)	3

**Table 2 cells-13-00696-t002:** The interaction of several proteins with actin fused with NLS significantly differs after irradiation with UV light. Welch’s *t*-test was performed, and the results are reported for significantly altered proteins. Q-value refers to *p*-values corrected for multiple testing, and log2-fold change refers to differences in the mean between the log2-transformed intensities of UV-treated cells vs. control.

Genes	*p*-Value	Q-Value	log2-Fold Change
GEMIN7	0.000386	0.044	−1.9
NUDT3	0.000135	0.027	−0.6
SC5D	0.00012	0.036	−1.8
SPTAN1	0.00042	0.045778	0.4
SPTBN1	9.41 × 10^−5^	0.054	0.5
VPS33A	0.000353	0.058667	−0.3
ZMIZ1	0.000495	0.0412	−2.8

## Data Availability

The materials and data are available upon reasonable request from the authors.

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
