# Peer review of "A Novel Interaction of Slug (SNAI2) and Nuclear Actin"

_cells, 2024, doi:10.3390/cells13080696_

Round 1
Reviewer 1 Report
Comments and Suggestions for Authors
The team of Zhuo and colleagues have identified in this study a novel interaction between the protein snail family transcriptional repressor 2 (slug) and b-actin, an interaction that is functionally relevant for cells to respond to DNA damage. Overall, the experiments in this study are well designed and carefully carried out with appropriate controls. The findings are interesting and an important contribution to the field.
One experiment that might add strength to the study is to test whether silencing of slug also changes the nuclear versus cytoplasmic levels of actin (in addition to impacting on the cytoplasmic actin organisation)?
Whilst testing the interaction of slug with gH2AX was tested after the induction of DNA damage with doxorubicin with and without expression of nuclear actin, the testing of the interaction after irradiation with UV light apparently has only been carried out with the expression of nuclear actin. Is there are specific reason why this has not been tested in the same format (without expression of nuclear actin)?
Regarding the statement in line 305, ‘this increase was even more pronounced after induction of nuclear actin (Figure 5a)’, has this been analysed for statistically significant differences? If not, this should be added.
Minor comments:
In line 27, branching should be included as one of the major functions of ABPs (e.g. by the ARP2/3 complex.
Under the Yeast two hybrid system in the Materials and Methods section, it should be specifically stated that the actin that was amplified is b-actin as this is important for interpreting the screen of interaction partners.
Author Response
One experiment that might add strength to the study is to test whether silencing of slug also changes the nuclear versus cytoplasmic levels of actin (in addition to impacting on the cytoplasmic actin organisation)?
This is indeed an interesting suggestion. However, it is technically not possible to image the high levels of cytoplasmic actin and the low levels of nuclear actin at the same time - at least not without manipulating nuclear actin levels.
Whilst testing the interaction of slug with gH2AX was tested after the induction of DNA damage with doxorubicin with and without expression of nuclear actin, the testing of the interaction after irradiation with UV light apparently has only been carried out with the expression of nuclear actin. Is there are specific reason why this has not been tested in the same format (without expression of nuclear actin)?
We thank the reviewer very much for this observation! Indeed, there was a mistake in our previous manuscript: the investigations with UV irradiation and the interaction of slug and H2AX were performed without nuclear overexpression of actin. The overexpression was used for the IP of nuclear actin an subsequent proteomics. We have now corrected this mistake.
Regarding the statement in line 305, ‘this increase was even more pronounced after induction of nuclear actin (Figure 5a)’, has this been analysed for statistically significant differences? If not, this should be added.
We have performed a statistical analysis, and now indicate the levels of statistical significance in the Figure legend.
Minor comments:
In line 27, branching should be included as one of the major functions of ABPs (e.g. by the ARP2/3 complex.
We have followed the reviewer´s suggestion.
Under the Yeast two hybrid system in the Materials and Methods section, it should be specifically stated that the actin that was amplified is b-actin as this is important for interpreting the screen of interaction partners.
We have now clarified this in the manuscript.
Reviewer 2 Report
Comments and Suggestions for Authors
Introduction of this Zhuo et al. paper is closed by "Here we describe the identification of an interaction between actin and slug, and characterize the impact on the DNA damage response." While the evidence supporting the actin-slug binding through yeast two-hybrid screening and antibody-mediated pulldown experiments is compelling, the paper falls short in substantiating the “the impact (of actin-slug interaction) on the DNA damage response”. This constitutes a major weakness in the manuscript. Additionally, a lack of a clear hypothesis on how the actin-slug interaction contributes to DNA damage repair makes it challenging to comprehend the relevance of each experiment in characterizing this interaction. The absence of adequate background information for each experiment further complicates understanding the experimental design, rendering the manuscript premature for publication in a reputable journal.
Major Comments:
1. The suggested functional significance of the actin-slug interaction in DNA damage repair, supported by increased slug-γH2AX colocalization due to elevated nuclear actin content, has two notable weaknesses. Firstly, the paper fails to demonstrate whether the increased slug-γH2AX colocalization is associated with improved DNA repair. Secondly, this observation occurs under an unphysiological condition of increased nuclear actin content, generating nuclear actin filaments artifactually. It is essential to establish a correlation between the extent of actin-dependent slug-γH2AX colocalization and DNA repair efficiency. The authors previously noted that the addition of the actin filament stabilizing agent chondramide B impairs DNA repair (ref 22). Is it possible that stabilizing actin filaments reduced the amount of functional nuclear actin in that experiment? If this point is established, one can either increase or decrease the concentration of functional nuclear actin experimentally, allowing one to investigate the correlation between the extent of actin-dependent slug-γH2AX colocalization and DNA repair efficiency.
2. The finding that silencing of slug increases actin aggregates in the cytoplasm, despite slug primarily localizing to the nucleus, is intriguing. However, the presented data is phenomenological and prompts mechanistic investigation. Increased actin expression due to slug's potential modulation of actin gene transcription or altered activity of actin binding proteins are two plausible mechanisms, and effects of slug silencing on actin expression should be examined experimentally.
3. Pulldown experiments (Table 2) aimed at identifying actin binding proteins using an anti-actin antibody have, unfortunately, diverted attention from the primary focus of the paper, as they failed to detect actin-slug interaction. Targeting slug-binding protein would be more appropriate for the scope of this paper, partly because actin has so many binding partners. If this set of data is to be maintained in this paper, please discuss why slug was not detected in this assay.
In Discussion it is stated “both (spectrins) interacted more with nuclear actin after UV irradiation” (line 385-386). Did Authors isolate nuclei and then obtained nucleoplasmic proteins for the pulldown assay? If, on the other hand, whole cell lysates were used, what is the basis to conclude that the spectrins interacted with nuclear actin, rather than cytoplasmic actin?
Provide explanations for terms such as "p-value," "q value," and "log2fold change" for non-specialists of proteomic analysis.
4. In Discussion, it is stated “silencing of slug stabilizes γH2AX foci, and thus damage resolution” (line 364) and “One could speculate that a ternary complex consisting of nuclear actin, slug, and γH2AX is formed during DNA damage repair.” (line 365-366). Are Authors implying that the role of slug is to resolve the ternary complex and promote the repair? If so, excessive slug-γH2AX colocalization caused by increased nuclear content (Figure 5) might inhibit, rather than promote, DNA repair. Coming back to major comment #1, measuring DNA repair in cells with increased nuclear actin would provide valuable insights into the role of slug in DNA repair.
Minor Comments:
1. Consider revising the title to avoid confusion around "slug" as a common noun, potentially leading to misinterpretation by readers unfamiliar with it as a transcription factor.
2. Replace the phrase "expression of nuclear actin" (lines 18, 312, 313, 363 and 374) with "expression of actin-NLS (or actin fused with NLS)" or "expression of Nuclear Actin-Chromobody TagGFP2" for accuracy.
3. “Actin was amplified from… (line 48)” should be changed to “Coding sequence of actin (or actin gene) was amplified from…”. Also, specify which actin gene was amplified.
4. In the legend for Figure 1, explain whether the nuclear membrane was disrupted during cell lysate preparation, as this information is essential for clarity.
5. Provide information on the duration of cell incubation after siRNA treatment before observing actin in Figure 2. Define "nt" in Figure 2 (a), and clarify the content of the left-most lane in Figure 2 (b). Additionally, explain the incubation duration with LatB or Jasp before measuring slug expression.
6. How were the number of actin aggregates counted in Figure 2 (a) and how were they distinguished from normal actin “aggregates” such as focal adhesions? I am asking this because there are apparently more than ten bright red spots in the ctrl cell image, whereas in the right quantification it is 1~2.
7. In Figure 2 (c), consider presenting absolute values of nuclear/cytoplasmic ratios for slug under different conditions for better understanding.
8. Provide basic information about the "Bliss Independence Value" in Figure 3, considering that most readers may be unfamiliar with this term.
9. In Figure 4, enhance clarity by displaying individual images of actin and slug at a larger magnification to better demonstrate weak colocalization.
10. Specify whether "repair time=0 h" refers to immediately after or before the addition of doxo. If it is the former, please include the data before the addition of doxo. Also, in the former case, is it implied that DNA was damaged as soon as doxo was added and the repair response started? In either case, consider replacing the horizontal axis label with “Time after the addition of doxorubicin (h)” if we do not know when the repair reaction started?
11. Clarify the visualization method for slug in Figure 4 and explain the nature of "pmCherry-C1 actin-3XNLS P2A" (this sounds like the name of the plasmid rather than the expressed gene or protein).
12. Provide explanations about RPA2 in the manuscript.
13. In Figure 5, specify whether "0 h" is immediately after the application of doxo. Also, if it is immediately after the application of doxo and the Pearson’s coefficient is elevated from ctrl, the increased colocalization between slug and γH2AX is a fairly rapid reaction. Please state how much time was actually elapsed between the application of doxo and the capture of the microscopic images. If 0 h is immediately before the addition of doxo, then please explain why the coefficient is different from ctrl.
14. Include statistical information to evaluate the significance of differences in Figure 5.
15. There are no curves in Figure 5. "red curve" and "blue curve" in the legend should be changed.
16. In Figure 6, LatB apparently increased the colocalization coefficient of slug and gH2AX. If the increases are statistically significant, please put this finding in the context of this paper. Did LatB exert similar effects when DNA was damaged by doxo?
Author Response
Major Comments:
- The suggested functional significance of the actin-slug interaction in DNA damage repair, supported by increased slug-γH2AX colocalization due to elevated nuclear actin content, has two notable weaknesses. Firstly, the paper fails to demonstrate whether the increased slug-γH2AX colocalization is associated with improved DNA repair. Secondly, this observation occurs under an unphysiological condition of increased nuclear actin content, generating nuclear actin filaments artifactually. It is essential to establish a correlation between the extent of actin-dependent slug-γH2AX colocalization and DNA repair efficiency. The authors previously noted that the addition of the actin filament stabilizing agent chondramide B impairs DNA repair (ref 22). Is it possible that stabilizing actin filaments reduced the amount of functional nuclear actin in that experiment? If this point is established, one can either increase or decrease the concentration of functional nuclear actin experimentally, allowing one to investigate the correlation between the extent of actin-dependent slug-γH2AX colocalization and DNA repair efficiency.
The reviewer raises two important points. First: We think that the occurrence of γH2AX foci and their correlation to the extent of DNA damage repair are well established in the literature, and that it is beyond the scope of our study to demonstrate a correlation between slug- γH2AX colocalization and functional DNA damage. We ask the reviewer to keep in mind that the focus of our study is a previously unknown interaction between slug and actin. Second: It is a well-known problem that nuclear actin can hardly be studied without unphysiologically altering nuclear actin levels. All available molecular tools have this shortcoming. We think it is accepted best practice to use two different tools in a comparative approach to at least rule out unspecific effects of the respective carrier. To this end, we used both, NLS-mcherry actin and NLS actin chromobodies. We now discuss the issue that increasing nuclear F-actin structures might bias the functional outcome.
- The finding that silencing of slug increases actin aggregates in the cytoplasm, despite slug primarily localizing to the nucleus, is intriguing. However, the presented data is phenomenological and prompts mechanistic investigation. Increased actin expression due to slug's potential modulation of actin gene transcription or altered activity of actin binding proteins are two plausible mechanisms, and effects of slug silencing on actin expression should be examined experimentally.
We agree that this finding is intriguing, and merits deeper mechanistic studies. However, this seems to be beyond the scope of this manuscript, where the focus lies on the nuclear aspects.
- Pulldown experiments (Table 2) aimed at identifying actin binding proteins using an anti-actin antibody have, unfortunately, diverted attention from the primary focus of the paper, as they failed to detect actin-slug interaction. Targeting slug-binding protein would be more appropriate for the scope of this paper, partly because actin has so many binding partners. If this set of data is to be maintained in this paper, please discuss why slug was not detected in this assay.
In Discussion it is stated “both (spectrins) interacted more with nuclear actin after UV irradiation” (line 385-386). Did Authors isolate nuclei and then obtained nucleoplasmic proteins for the pulldown assay? If, on the other hand, whole cell lysates were used, what is the basis to conclude that the spectrins interacted with nuclear actin, rather than cytoplasmic actin?
We apologize for the missing explanation. For enriching nuclear actin for the pulldown, we overexpressed mcherry labelled actin with multiple nuclear localization sequences (the same plasmid as for the imaging), and precipitated with an anti-mcherry antibody. We have stated this now in the Methods section (under the heading “Immunoprecipitation”. We were disappointed when we did not find slug in the list of proteins bound to nuclear actin. However, it has to be kept in mind that mass spectrometry has sensitivity limitations. Obviously, the abundance of slug was too low for detection. We added this point to the discussion.
Provide explanations for terms such as "p-value," "q value," and "log2fold change" for non-specialists of proteomic analysis.
We have added some explanations to the legend of Table 2.
- In Discussion, it is stated “silencing of slug stabilizes γH2AX foci, and thus damage resolution” (line 364) and “One could speculate that a ternary complex consisting of nuclear actin, slug, and γH2AX is formed during DNA damage repair.” (line 365-366). Are Authors implying that the role of slug is to resolve the ternary complex and promote the repair? If so, excessive slug-γH2AX colocalization caused by increased nuclear content (Figure 5) might inhibit, rather than promote, DNA repair. Coming back to major comment #1, measuring DNA repair in cells with increased nuclear actin would provide valuable insights into the role of slug in DNA repair.
At the current stage, we can only speculate on this. These questions will be addressed in follow-up projects.
Minor Comments:
- Consider revising the title to avoid confusion around "slug" as a common noun, potentially leading to misinterpretation by readers unfamiliar with it as a transcription factor.
We have now added the alias SNAI2 to the title to avoid confusion.
- Replace the phrase "expression of nuclear actin" (lines 18, 312, 313, 363 and 374) with "expression of actin-NLS (or actin fused with NLS)" or "expression of Nuclear Actin-Chromobody TagGFP2" for accuracy.
We agree, and have replaced the phrase wherever appropriate.
- “Actin was amplified from… (line 48)” should be changed to “Coding sequence of actin (or actin gene) was amplified from…”. Also, specify which actin gene was amplified.
We have adapted the passage and specified beta actin.
- In the legend for Figure 1, explain whether the nuclear membrane was disrupted during cell lysate preparation, as this information is essential for clarity.
For cell lysis we used RIPA buffer. There is no extra step for lysis of the nuclear membrane. We added the information that RIPA buffer lyses both, cell membrane and nuclear membrane to the Methods part.
- Provide information on the duration of cell incubation after siRNA treatment before observing actin in Figure 2. Define "nt" in Figure 2 (a), and clarify the content of the left-most lane in Figure 2 (b). Additionally, explain the incubation duration with LatB or Jasp before measuring slug expression.
The information on the duration of siRNA treatment was already indicated in the Methods section (chapter “Transfection”) as 24h in the original manuscript. We have now repeated this information in the new panel a of Fig.2. nt, the content of the left-most lane (size marker) and the incubation time (24 h) of the compounds is now indicated.
- How were the number of actin aggregates counted in Figure 2 (a) and how were they distinguished from normal actin “aggregates” such as focal adhesions? I am asking this because there are apparently more than ten bright red spots in the ctrl cell image, whereas in the right quantification it is 1~2.
The number of the aggregates was quantified automatically using ImageJ based on the size and shape of the particles (the setting of size: 0.03-5.00, circularity: 0.2-1.00). These settings were selected to represent the impression by eye in the best un-biased way. This is now also stated in the Methods section (Confocal microscopy).
- In Figure 2 (c), consider presenting absolute values of nuclear/cytoplasmic ratios for slug under different conditions for better understanding.
The densitometric data have to be normalized to equal loading of the membrane anyway. We prefer not to raise the impression of “absolute” values here.
- Provide basic information about the "Bliss Independence Value" in Figure 3, considering that most readers may be unfamiliar with this term.
We have now added some information in the legend to Fig. 3.
- In Figure 4, enhance clarity by displaying individual images of actin and slug at a larger magnification to better demonstrate weak colocalization.
The resolution shown in the Figure panels is the highest possible. Zooming in would not add optical information.
- Specify whether "repair time=0 h" refers to immediately after or before the addition of doxo. If it is the former, please include the data before the addition of doxo. Also, in the former case, is it implied that DNA was damaged as soon as doxo was added and the repair response started? In either case, consider replacing the horizontal axis label with “Time after the addition of doxorubicin (h)” if we do not know when the repair reaction started?
This terminology is indeed misleading. Doxo was added for two hours, and then removed again. The medium change was set as repair time 0. We now use the term “time after medium change” in th Figure.
- Clarify the visualization method for slug in Figure 4 and explain the nature of "pmCherry-C1 actin-3XNLS P2A" (this sounds like the name of the plasmid rather than the expressed gene or protein).
Slug was visualized by indirect immunofluorescence. This, and the nature of the fusion construct is now indicated in the legend to the figure.
- Provide explanations about RPA2 in the manuscript.
We have now added Fig. panel 6b to show the interaction between slug and RPA2. All in all, RPA2 turned out not to be prominently involved in the main findings of our manuscript. We, thus, would not like to discuss this protein more pronounced as in the previous version of the manuscript.
- In Figure 5, specify whether "0 h" is immediately after the application of doxo. Also, if it is immediately after the application of doxo and the Pearson’s coefficient is elevated from ctrl, the increased colocalization between slug and γH2AX is a fairly rapid reaction. Please state how much time was actually elapsed between the application of doxo and the capture of the microscopic images. If 0 h is immediately before the addition of doxo, then please explain why the coefficient is different from ctrl.
Please see response to point 10.
- Include statistical information to evaluate the significance of differences in Figure 5.
We have now included the statistical information.
- There are no curves in Figure 5. "red curve" and "blue curve" in the legend should be changed.
We have changed the legend accordingly.
- In Figure 6, LatB apparently increased the colocalization coefficient of slug and gH2AX. If the increases are statistically significant, please put this finding in the context of this paper. Did LatB exert similar effects when DNA was damaged by doxo?
We now added the statistical data to the Figure – there is no significant difference between these two groups. We do not have this data set for the doxo treatment.
Reviewer 3 Report
Comments and Suggestions for Authors
This is a study to investigate a novel interaction between nuclear actin and the transcription factor Slug. The authors start off by describing their identification of this interaction via a screen for binding partners of actin using yeast-two-hybrid. The authors next validate this interaction using pulldowns and further explore the effect of loss of Slug on the architecture of the F-actin cytoskeleton. Finally, the authors explore how the interaction of Slug with actin and other known binding partners changes during DNA damage.
The experiments are relatively well-done and the figures are easy to understand and appropriate for the content.
I have a few suggestions:
1) In figure 1, the authors show that IP of actin pulls down slug and vice versa. For each pulldown done, the figure should show a blot of both components. This will allow the audience to gage how efficient the pulldown was…
2) In figure 2, the authors knockdown slug and see an effect of the actin cytoskeleton. The efficiency of slug knockdown is not shown however. They need to blot for slug in their knockdown and/or show IF of slug to ensure the knockdown efficiency. Also, a rescue experiment putting a refractory version of slug back into the cells would help convince the reader of their claims.
3) In figure 4, the colocalization of slug and nuclear actin is not very convincing. The IF basically doesn’t allow you to make any conclusions of the interaction, since there are no controls. Two diffuse molecules in the nucleus are going to overlap. Would you see the same thing with mCherry-NLS and slug? The authors should at least use proximity ligation assays (with controls) to provide evidence for any interaction.
Author Response
1) In figure 1, the authors show that IP of actin pulls down slug and vice versa. For each pulldown done, the figure should show a blot of both components. This will allow the audience to gage how efficient the pulldown was…
We have followed the reviewer´s suggestion and adapted Figure 1 accordingly. We have now also added the lanes with the flow-through to indicate the total amount of the proteins in the respective samples.
2) In figure 2, the authors knockdown slug and see an effect of the actin cytoskeleton. The efficiency of slug knockdown is not shown however. They need to blot for slug in their knockdown and/or show IF of slug to ensure the knockdown efficiency. Also, a rescue experiment putting a refractory version of slug back into the cells would help convince the reader of their claims.
We have now added blots for slug and their quantitative evaluation after silencing in Fig. 2a. We see a clear downregulation by about 80% as compared to the basal level. Performing a rescue experiment would be a great idea. However, since the focus of the manuscript was more on the nuclear/DNA damage side of the story, we have decided against these additional experiments.
3) In figure 4, the colocalization of slug and nuclear actin is not very convincing. The IF basically doesn’t allow you to make any conclusions of the interaction, since there are no controls. Two diffuse molecules in the nucleus are going to overlap. Would you see the same thing with mCherry-NLS and slug? The authors should at least use proximity ligation assays (with controls) to provide evidence for any interaction.
We agree that IF does technically not allow for interaction studies, due to the limit of lateral resolution of approx. 250 nm – which is a lot at the scale of protein sizes. FRET experiments or proximity ligation assays would be the methods of choice, but establishing these assays and repeating the key experiments is beyond the scope of this project. In order to avoid a wrong impression on the readers, we have now rephrased the respective parts in the manuscript, and replaced the phrase “interact” by “spatially correlate”. As we have clear quantitative effects on the colocalizations both with the mCherry NLS actin (the reviewer seems to have missed these data, Fig. 4a) and with the structurally unrelated nuclear actin chromobody (Fig. 4b), we are quite confident, that our data corroborate the message of our corrected text.
Round 2
Reviewer 2 Report
Comments and Suggestions for Authors
Zhuo et al. adequately addressed all my concerns to the original version. Ambiguities in experimental procedures which hampered interpretation of the data are resolved. The possible artefacts due to the use of NLS-actin and NLS-actin chromobody are explicitly mentioned in Discussions. I am happy with this revised manuscript.
Reviewer 3 Report
Comments and Suggestions for Authors
The authors adequately addressed my concerns.